# OpenReview forum: "Just Enough Shifts: Mitigating Over-Refusal in Aligned Language Models with Targeted Representation Fine-Tuning"
_ICML.cc/2025/Conference — ICML 2025 poster_

### Official Review · Reviewer_eEU9 · 2025-03-06

**Overall Recommendation:** 2

**Summary:**

This paper addresses the over-refusal problem in aligned LLMs that unnecessarily reject benign user prompts. The authors identify specific layers whose latent representations best distinguish between benign and malicious prompts, then selectively adjust embeddings to move prompts "just enough" from rejection to acceptance. Their approach uses a proxy that measures refusal contribution through query projection onto the refusal vector, deriving the shift from a locally linear approximation of the refusal boundary. Experiments across three models demonstrate higher compliance rates than fine-tuning while maintaining safety scores and general functionality.

**Claims And Evidence:**

Yes

**Essential References Not Discussed:**

No

**Experimental Designs Or Analyses:**

Yes

**Methods And Evaluation Criteria:**

Yes

**Other Comments Or Suggestions:**

The presentation can be improved. For example, in Figure 1, there is too much space inside the quote box and the text is not centered.

**Other Strengths And Weaknesses:**

Strength:
1. One of the key novelties of this paper lies in the proposal of "just enough" shift. The paper gives one motivation as uniform shift leads to gebbrish generation.
2. The method targets specific layers for adjustment rather than modifying the entire model,  requires minimal compute compared to full model retraining
3. Results show improved compliance rates compared to fine-tuning approaches while maintaining safety scores and general model functionality

Weakness:
1. The proposed method is not applicable to black-box models.
2. The generalization of the learned shift remains untested. Training on some datasets and evaluating on non-overlapping datasets would provide insight into its robustness.
3. The evaluation is limited in scope—only two architectures are tested, and activation-based baselines are only evaluated on one model.

**Questions For Authors:**

1. Can you clarify what you mean by pseudo-harmful prompts?
2. How does the learned shift generalize to unseen datasets? Have you considered training on some datasets and evaluating on others?
3. Why were activation-based baselines evaluated on only one model? Would broader comparisons strengthen the claims?
4. Given that the method is not applicable to black-box models, do you see potential adaptations that could make it usable in such settings?

**Relation To Broader Scientific Literature:**

The paper mitigates the over-refusal issue of LLMs while maintaining a high safety score and general capability, enhancing the usability of LLMs.

**Theoretical Claims:**

N/A

---

> ### Author Rebuttal · Authors · 2025-04-01
>
> Thank you for your insightful feedback. We have provided our responses below, and we hope they clarify the points you raised. If our responses have adequately addressed your initial concerns, we would be grateful if you would consider adjusting your evaluation accordingly.
>
> ## (A)
>
> We define pseudo-harmful prompts in line 39 of the Introduction- they are queries which appear potentially harmful but are benign in nature.
>
> ## (B)
>
> We address the question of generalization to unseen datasets through our experimental setup. We train our model on 25 examples each from XSTest, Scope, ORBench, and PHTest, and then evaluate on the corresponding held-out partitions of those datasets. In addition, we include OKTest as an entirely unseen benchmark to assess how well our learned shift extends beyond the trained domains. The results are summarized in Table 1.
>
> To further address concerns about out-of-domain generalization, we also conduct a supplementary experiment in which training is restricted to only 25 queries from XSTest and ORBench along with the harmful and harmless queries from Hexphi and UltraChat. As shown in the experiments below, the model retains its out-of-distribution (OOD) robustness even when we vary the over-refusal distributions used during training.
>
> | Method | XS Test CR (In-Dist) | SCOPE CR (OOD)| Orbench CR (OOD)| Phtest CR (OOD)| OKTest CR (OOD)| Avg OR Compliance Rate| Advbench Safety Score| Tradeoff Score |
> |-|-|-|-|-|-|-|-|-|
> | ACTOR (25 XS Test) | 96.00| 90.39| 73.09| 95.03| 94.00| 89.70| 99.03| 94.37 |
>
> | Method | XS Test CR (OOD)| SCOPE CR(OOD)| Orbench CR (In-Dist)| Phtest CR(OOD)| OKTest CR (OOD) | Avg OR Compliance Rate| Advbench Safety Score| Tradeoff Score |
> |-|-|-|-|-|-|-|-|-|
> | ACTOR (25 ORB-H) | 94.67| 90.73| 73.91| 96.05| 94.33| 89.94| 98.85| 94.40 |
>
> ## (C)
>
> We agree that broader comparisons can further strengthen our work. We have now incorporated comparisons against multiple baselines.  First, we present the following table discussing the features of all existing approaches and their comparison with ACTOR.
>
> | Method| Mitigation of Over-Refusal | Train Time/ Inference Time | Robustness |
> |-|-|-|-|
> | Self-CD| High| **Inference Time** - Requires 2 generations for the same input | -|
> | DRO| Poor| **Train Time** - Introduces external parameters | -|
> | Safety Patching| High | **Train Time** - Requires training twice| - |
> | Safe-Decoding| Poor | **Train Time** - Requires 2 generations for the same input   | -|
> | SCANS| High| Inference Time| Low|
> | Surgical| High| Inference Time| Low |
> | ACTOR| High| Train Time | High|
>
> 1. **Results for Llama-2-7b-chat-hf**
>
> | Method| XS Test CR | SCOPE CR| Orbench CR | Phtest CR| OKTest CR) | Avg OR CR | Advbench SS | Tradeoff Score |
> |--|-|-|-|-|-|-|-|-|
> | Default| 80 | 52.61| 29.45 | 69.6 | 76 | 61.53 | 99.62| 80.58|
> | Safe-Decoding | 29.12| 15.32  | 7.45 | 26.03  | 45.24 | 24.63 | 100 | 62.32 |
> | DRO| 58 | 21.25  | 14.11 | 62.22 | 76 | 46.32| 100| 73.16|
> | Self CD| 90.67| 80.94| 61.94| 87.41| 92| 82.59| 95.77| 89.18|
> | SCANs| 95.33| 76.72| 40.52| 90.44| 99 | 80.40| 99.23| 89.82|
> | Surgical | 90.67| 89.38| 69.16| 93.42| 89.33| 86.39| 99.42| 92.90 |
> | Ours| 95.33| 91.57| 76.28| 96.86| 93.67 | **90.74** |99.03| **94.88**|
>
>
> 2. **Results for Gemma-7b-it**
>
> | Method| XS Test CR | SCOPE CR | Orbench CR | Phtest CR| OKTest CR | Avg OR CR | Advbench SS | Tradeoff Score |
> |-|-|---|--|--|-|--|--|-|
> | Default | 72.01| 58.18 | 65.71 | 88.92 | 74.00 | 71.76 | 94.00 | 82.88 |
> | Safe-Decoding | 32.12| 19.43 | 8.32| 38.21 | 40.34| 27.68| 98.32| 63.00|
> | DRO| 52.04| 44.92| 58.39| 75.01| 71.28| 60.33| 97.78| 79.06 |
> | Self CD| 78.00| 64.75| 74.20| 88.08| 73.00| 75.61| 87.12 | 81.36|
> | SCANs | 56.66 | 56.15| 70.87| 80.12| 53.66| 63.49 | 93.65| 78.57|
> | Surgical| 76.67 | 61.20 | 74.20 | 89.72| 76.33 | 75.62| 90.96| 83.29|
> | Ours| 79.33 | 62.73| 73.83| 91.15| 78.00| **77.01**| 92.5| **84.75**|
>
> Our results continue to show that ACTOR outperforms these broader baselines in reducing over-refusals while maintaining safety scores.
>
> ## (D)
>
> We acknowledge that our approach relies on access to model internals, which may be unavailable for users of closed-source or black-box systems. However, we do not view this as a fundamental limitation. In practice, closed-source model developers do have full access to these internals and can adopt our method if they choose. Moreover, the ability to examine the model’s internal representation space provides critical insights for understanding and mitigating over-refusals—insights that purely API-level or prompt-based methods cannot readily capture.

---

### Official Review · Reviewer_kLYp · 2025-03-10

**Overall Recommendation:** 3

**Summary:**

The paper proposes a fine-tuning based method to solve the over-refusal problem encountered by many LLMs. The method first tries to extract an over-refusal vector from the models using different prompts and then it tries to steer the model towards the embedding as defined in equation 9. The overall performance is strong as measured on various benchmarks.

**Claims And Evidence:**

Yes

**Essential References Not Discussed:**

N/A

**Experimental Designs Or Analyses:**

Yes, the experiment design is valid.

**Methods And Evaluation Criteria:**

Yes

**Other Comments Or Suggestions:**

N/A

**Other Strengths And Weaknesses:**

## Strength

1. The paper is well written and easy to follow
2. The proposed method is simple and effective
3. After fine-tuning, the model's original performance is well-preserved.



## Weakness
1. the proposed method is very close to [1] except with a fine-tuning stage, this makes the contribution less significant. And [1] is not discussed extensively in the work.
2. lack of comparison with baselines, although SafeDecoding or DRO may show lower performances, but it's better to have them in the table for comparison. Are they better or worse than SFT?
3. lack of ablation study on the number of calibration samples, how is 210 and 25 selected for the training dataset, how does it affect the downstream performances if you change them?
4. all the results are reported at a high level, but most benchmarks have provided specific categories for refusal. what are the performance on specific categories in the benchmarks. Currently there doesn't seem to be such analysis

[1] Surgical, Cheap, and Flexible: Mitigating False Refusal in Language Models via Single Vector Ablation

**Questions For Authors:**

1. can the author explain the difference between the current work and  [1], looks like it's built upon [1] with a little bit fine-tuning.

[1] Surgical, Cheap, and Flexible: Mitigating False Refusal in Language Models via Single Vector Ablation

**Relation To Broader Scientific Literature:**

The paper contributes a method to greatly reduce the over-refusal behaviors without affect its original performance. So it will be generally useful to trading off between being safe and helpful.

**Theoretical Claims:**

The theoretical claim is not rigorous since the problem cannot be exactly defined. Thus most of the conclusion in the paper is empirical.

---

> ### Author Rebuttal · Authors · 2025-04-01
>
> Thank you for your insightful feedback. We have provided our responses below, and we hope they clarify the points you raised. If our responses have adequately addressed your initial concerns, we would be grateful if you would consider adjusting your evaluation accordingly.
>
> ## (A)
>
> Our approach differs fundamentally from the single‐vector “surgical” method [1], which computes a “false‐refusal” vector from a set of harmless, harmful, and pseudo‐harmful queries. We observe that simply switching the pseudo‐harmful dataset from ORB-H to OKTest led to substantial fluctuations in performance.
>
> | Method | OR Compliance Rate |
> |-|-|
> | Surgical (ORB-H) | 86.39 |
> | Surgical (OKTest)  | 63.88 |
>
> These fluctuations arise because the method bakes a **single fixed vector** directly into the model’s weights and applies it uniformly to every query—a “one‐size‐fits‐all” mechanism. Consequently, the model’s efficacy heavily depends on the specific distribution from which the vector is extracted, rendering it brittle under distributional shifts. shifts. We explore a natural extension of this approach in the fine‐tuning setting under the subsubsection “Would Uniform Shifts Work?” (Sec 3.2). We show that a single uniform vector shift leads to **destructive fine‐tuning**, with more discussion in Appendix B.
>
> Instead of a single fixed vector, ACTOR repeatedly updates both the model parameters and the refusal direction during fine‐tuning. Mathematically, rather than enforcing a constant scaling factor, ACTOR’s loss function promotes an **“individualized” or “just enough” shift**—proportional to each query’s **projection onto the refusal direction**—so that the model can adapt its internal representation on a query‐by‐query basis via **minimal intervention**. We discuss these design choices in more detail in Section 3. This iterative, dynamic mechanism makes ACTOR **robust**: instead of relying on a single axis of correction, it exploits the full capacity of the model’s internal activation space to handle diverse data distributions.
>
> To further validate this robustness, we computed the initial refusal direction for ACTOR using three different harmful data distributions as shown in Fig 5 (Sec 4.2) and the table below. ACTOR consistently **maintains strong performance**, even when the refusal direction is computed from different data sources.
>
> | Datasets Used for Refusal Direction | Method | Avg OR Compliance Rate  |Advbench Safety Score|
> |--|-|-|-|
> | D_harmful = Hexphi  | ACTOR  | 90.01 | 99.03  |
> |  D_harmful = BeaverTails | ACTOR  | 89.84| 98.95 |
> | D_harmful = Malicious| ACTOR  | 89.94  | 98.85 |
>
>
> We hope this clarifies why our method is not merely an extension of [1] augmented with fine‐tuning. Our design deliberately addresses the pitfalls of a single‐vector solution and provides a more adaptive, reliable alignment strategy.
>
> ## (B)
>
> **Baselines**- Kindly refer to our rebuttal to Reviewer eEU9 (C). Both SafeDecoding and DRO suffer from a major over-refusal problem and perform worse than SFT.
>
> ## (C)
>
> The 210 benign examples are randomly sampled from the 7 categories of the UltraChat Dataset (n=30). The 25 queries from XSTest, SCOPE, OR-Bench-Hard-1k, PhTest benchmarks are also sampled randomly with their held-out versions used for evaluation and OKTest serving as an OOD dataset.
>
> We experiment with 2 additional settings where we set **n=15 and 50** for UltraChat-
>
> | Method| XS Test CR | SCOPE CR | Orbench CR | Phtest CR | OKTest CR | Avg CR | Advbench SS | Tradeoff Score |
> |-|-|-|-|-|-|-|-|--|
> | ACTOR (**n=15**)  | 94.67 | 89.88 | 72.76| 96.21 | 94.00| 89.50 | 98.85 | 94.17 |
> | ACTOR (**n=50**)  | 95.00| 90.39 | 73.09 | 96.21| 93.67| 89.60| 99.03| 94.31|
>
> Similarly, we also conducted additional ablation experiments by selecting **10** and **50** random over-refusal queries from the abovementioned datasets-
>
> | Method | XS Test CR | SCOPE CR | Orbench CR | Phtest CR | OKTest CR) | Avg CR| Advbench SS | Tradeoff Score |
> |-|-|-|-|-|-|-|-|-|
> | ACTOR (**n=10**)| 95.33 | 90.05| 73.01| 95.90  | 94.00| 89.65| 99.00 | 94.32 |
> | ACTOR (**n=50**)  | 95.33| 91.91| 74.98 | 96.81| 93.67| 90.54| 98.75| 94.65|
>
> We also want to highlight that our method remains effective even under **low data budgets**. As shown in Fig 4, using **only 25 over‐refusal queries** during training yields performance gains that **surpass SFT trained on 100 over‐refusal queries**, underscoring the **data efficiency** of our approach.
>
> ## (D)
>
> We agree that adding analysis on specific categories would make the paper more rich. XSTest, ORBench-Hard-1k and SCOPE are the benchmarks with such categories included. We show the Compliance Rates on these categories before and after intervention with ACTOR [here](https://limewire.com/d/HGAko#HqHk2E8Qst).
>
> [1] Surgical, Cheap, and Flexible: Mitigating False Refusal in Language Models via Single Vector Ablation

---

### Official Review · Reviewer_b27H · 2025-03-13

**Overall Recommendation:** 4

**Summary:**

Language Models (LMs) must balance refusing unsafe prompts while complying with benign ones. Despite safety training, LMs often refuse benign prompts that contain spurious correlations with harmful ones, a behavior known as over-refusal. This paper introduces ACTOR, a technique inspired by representation engineering. This technique involves first identifying a refusal direction in activation space and then fine-tuning a target layer to minimally shift model activations along this direction based on labels indicating the harmfulness of the training query. Empirical evidence shows that this representation-based technique is more data-efficient and task-performant than traditional Supervised Fine-Tuning (SFT).

**Claims And Evidence:**

The paper provides strong empirical evidence for its claims. Beyond studying over-refusal directly, the paper benefits from comparing against test-time steering interventions which are, at present, a popular approach in the literature.

However, the paper focuses on single-turn harmful prompts. While this is a common evaluation setup in the literature, it is unclear whether the robustness generalizes to challenging multi-turn attacks [1, 2] which may represent a more realistic threat model [3]. I suggest that the authors either include multi-turn experiments or acknowledge this gap as a potential limitation in the generalization of their results.

[1] - Russinovich, M., Salem, A., & Eldan, R. (2024). Great, Now Write an Article About That: The Crescendo Multi-Turn LLM Jailbreak Attack. ArXiv, abs/2404.01833.

[2] - Li, N., Han, Z., Steneker, I., Primack, W.E., Goodside, R., Zhang, H., Wang, Z., Menghini, C., & Yue, S. (2024). LLM Defenses Are Not Robust to Multi-Turn Human Jailbreaks Yet. ArXiv, abs/2408.15221.

[3] - Haider, E., Perez-Becker, D., Portet, T., Madan, P., Garg, A., Majercak, D., Wen, W., Kim, D., Yang, Z., Zhang, J., Sharma, H., Bullwinkel, B., Pouliot, M., Minnich, A., Chawla, S., Herrera, S., Warreth, S., Engler, M., Lopez, G., Chikanov, N., Dheekonda, R.S., Jagdagdorj, B., Lutz, R., Lundeen, R., Westerhoff, T., Bryan, P., Seifert, C., Kumar, R.S., Berkley, A., & Kessler, A. (2024). Phi-3 Safety Post-Training: Aligning Language Models with a "Break-Fix" Cycle. ArXiv, abs/2407.13833.

**Essential References Not Discussed:**

There are no obvious missing references that aren't considered concurrent work.

**Experimental Designs Or Analyses:**

I did check the experiment design and read the papers for the leveraged benchmarks. This experiment design is in line with the existing literature. The authors acknowledge that using a variety of refusal/safety benchmarks can have the confounding factor of variance in the labeling policies of the benchmark authors.

**Methods And Evaluation Criteria:**

The evaluation metrics and datasets fit the proposed research questions. The paper especially benefits from studying out-of-distribution (OOD) robustness, data efficiency, and multi-turn overall performance.

**Other Comments Or Suggestions:**

There is a typo on line 330.

**Other Strengths And Weaknesses:**

Strength: This work is well-written, especially the section describing the theoritical motivations of the technique.

Weakness: The paper seems to use a custom GPT-4o prompt for refusal and harm classification. There are existing classifiers int he literature for this task such as HarmBench [1], LlamaGuard [2], and Wild Guard [3]. Using a custom prompt makes comparisions across papers more difficult.

[1] - Mazeika, M., Phan, L., Yin, X., Zou, A., Wang, Z., Mu, N., Sakhaee, E., Li, N., Basart, S., Li, B., Forsyth, D., & Hendrycks, D. (2024). HarmBench: A Standardized Evaluation Framework for Automated Red Teaming and Robust Refusal. ArXiv, abs/2402.04249.

[2] - Inan, H., Upasani, K., Chi, J., Rungta, R., Iyer, K., Mao, Y., Tontchev, M., Hu, Q., Fuller, B., Testuggine, D., & Khabsa, M. (2023). Llama Guard: LLM-based Input-Output Safeguard for Human-AI Conversations. ArXiv, abs/2312.06674.

[3] - Han, S., Rao, K., Ettinger, A., Jiang, L., Lin, B.Y., Lambert, N., Choi, Y., & Dziri, N. (2024). WildGuard: Open One-Stop Moderation Tools for Safety Risks, Jailbreaks, and Refusals of LLMs. ArXiv, abs/2406.18495.

**Questions For Authors:**

My understanding is that the authors perform full-parameter fine-tuning for the target layer. Have the authors looked into parameter-efficient techniques like LoRA? Successful parameter-efficient experiments can further demonstrate the effectiveness of the technique and allow experimentation with even larger models.

**Relation To Broader Scientific Literature:**

Over-refusal is a prominent challenge in modern LM safety training. This work makes a valuable contribution by showing that focusing on optimizing against internal representations during train-time can outperform traditional fine-tuning as well as dynamic test-time steering techniques.

**Theoretical Claims:**

The paper motivates the proposed technique with theoretical claims regarding the geometry of refusal. The paper cautions that the theoretical intuition serves as a useful motivation for the technique but does not serve as proof in its own right and is thus not to be considered a core contribution of the work. I did not attempt to prove these claims and instead rely on the empirical evidence as support for these theoretical claims.

---

> ### Author Rebuttal · Authors · 2025-04-01
>
> We appreciate your thoughtful remarks and the positive rating you assigned to our paper.  Below, you'll find our responses, which we hope clarify the points you raised.
>
> ## (A)
>
> While multi‐turn attacks indeed pose a more realistic challenge, there are currently no established benchmarks specifically designed for multi‐turn over‐refusal scenarios. Evaluating both multi‐turn over-refusal and multi‐turn safety, as well as understanding how they intersect, remains an open research problem. We therefore leave this exploration to future work, and we appreciate your suggestion to further examine the robustness of our method in more complex dialog settings.
>
> ## (B)
>
> We found that existing classifiers such as HarmBench, LlamaGuard, and WildGuard can be overly conservative for the pseudo-harmful queries used in our benchmarks, often flagging them as harmful. For example- **LlamaGuard classifies 33% of the total queries from ORBench-Hard-1k as harmful**.
>
> Earlier approaches [1][2]  in our line of work include human evaluation and string matching for evaluation. To achieve a balance between context‐awareness, reproducibility, and scalability we opted for an LLM judge with a carefully crafted system prompt—an approach used extensively in related literature [3]. We provide all relevant prompts and details in Appendix E, allowing others to replicate our setup and compare results more directly.
>
> ## (C)
>
> Since our current approach only fine-tunes a single layer—already providing a relatively efficient setup—we consider integrating LoRA into our design as an exciting avenue for further increasing parameter efficiency and performance even on larger models.
>
> Thanks again for the valuable suggestions.
>
> [1] Cao, Zouying, Yifei Yang, and Hai Zhao. "Nothing in excess: Mitigating the exaggerated safety for llms via safety-conscious activation steering." arXiv preprint arXiv:2408.11491 (2024).
>
> [2] Wang, Xinpeng, et al. "Surgical, Cheap, and Flexible: Mitigating False Refusal in Language Models via Single Vector Ablation." arXiv preprint arXiv:2410.03415 (2024).
>
> [3] Qi, Xiangyu, et al. "Fine-tuning aligned language models compromises safety, even when users do not intend to!." arXiv preprint arXiv:2310.03693 (2023).

---

> > ### Comment · Reviewer_b27H · 2025-04-02
> >
> > Thank you for responding to my questions. My score remains unchanged.

---

> > > ### Author Response · Authors · 2025-04-06
> > >
> > > Dear Reviewer b27H,
> > >
> > > We are happy to hear that our responses have addressed your concerns and questions. We appreciate you taking the time to read our rebuttal and adjust your evaluation accordingly. We will incorporate all the clarifications, additional experimental results, and suggested modifications discussed during the rebuttal into our revised version. Thank you once again for your valuable, constructive feedback and for your consideration.
> > >
> > > Best regards,
> > >
> > > Authors

---

### Official Review · Reviewer_Fyfs · 2025-03-14

**Overall Recommendation:** 3

**Summary:**

This paper focuses on addressing the over-refusal issue in aligned LLMs. The proposed technique, ACTOR, leverages internal activation for fine-tuning a single layer of the model to reduce the over-refusal rate.


## update after rebuttal

Thanks for the authors' response, which addresses most of my concerns.

**Claims And Evidence:**

Overall, most evidence is clearly presented. However, I have some concerns regarding the limitations of existing work claimed in Section 2, which I did not find references or evidence to support, for example (not limited to)

> These
inference-time solutions, while computationally efficient,
are highly sensitive to initial data distributions, leading to
inconsistent performance across different contexts. Additionally, these approaches offer a one-size-fits-all solution
and typically do not provide differentiated treatment for various types of queries.

**Essential References Not Discussed:**

NA

**Experimental Designs Or Analyses:**

Mostly comprehensive, subject to the coverage of models, datasets, and ablation studies. However, the experiment only includes SFT and SCANS as baselines, missing comparison with other broad existing methods discussed in this paper. There is also a lack of an ablation study on the selection of the fine-tuning layer.

**Methods And Evaluation Criteria:**

The proposed method, using representation vectors for fine-tuning, intuitively makes sense. However, it seems directly applying representation fine-tuning methods in this scenario, so I think the novelty can be further justified.

**Other Comments Or Suggestions:**

See above sections.

**Other Strengths And Weaknesses:**

Some figures/tables are overly large/small. A better format is appreciated.

**Questions For Authors:**

See above sections.

**Relation To Broader Scientific Literature:**

Addressing the over-refusal problem could benefit the balance between safety and utility of LLMs.

**Theoretical Claims:**

The theoretical analysis is correct but overly simplified and straightforward. Moreover, it does not align with other main claims made in this paper, e.g. why ”Just Enough” Shift does not harm natural performance or why a single layer is sufficient to achieve the fine-tuning.

---

> ### Author Rebuttal · Authors · 2025-04-01
>
> Thank you for your thoughtful feedback. Below, you'll find our detailed responses, which we hope clarify the points you raised.
>
> ## (A)
>
> We appreciate the opportunity to clarify the basis of our claims in Sec 2. While we had experimental results (Sec 4) that support these points, we realize they were not explicitly cited. We will revise the manuscript to better link these findings to our claims.
>
> 1. We claim that SCANS and the “surgical” method [1]—are highly sensitive to the specific datasets used to derive their refusal vectors. In Figure 5, we show that SCANS exhibits significant performance fluctuations when we vary the harmful datasets used. Likewise [1] also exhibits similar variance when switching among different pseudo-harmful datasets.
>
> | Method| OR Compliance Rate |
> |-|-|
> | Surgical (ORB-H) | 86.39|
> | Surgical (OKTest)  | 63.88 |
>
> 2. We also claimed that SCANS and [1] offer a uniform, one-size-fits-all adjustment to the model, applying the same refusal vector—regardless of each query’s unique activation patterns. This approach can lead to unnecessary or insufficient corrections, as each query requires a different level of ablation to ensure compliance (Sec 3.2).
>
> **ACTOR’s Adaptive Mechanism**: In contrast, our proposed approach employs a **dynamic training objective**- moving ahead from specific task vectors. ACTOR repeatedly updates both the model parameters and the refusal direction during fine-tuning, enabling an **input-dependent shift** proportional to each query’s projection onto the refusal direction. This underpins ACTOR’s robustness; it is not limited to a single axis correction but leverages the full capacity of the model’s internal activation space to maintain robust performance even under distributional shifts (Fig 5). As shown by our experimental results in the paper (Tables 1–4) and the addition of new baselines (Reviewer eEU9 response (C))- ACTOR outperforms all baselines in balancing both compliance and safety. We have also included a table comparing existing methods with ACTOR to further support our claims on their limitations in our response to Reviewer eEU9 (C).
>
> ## (B)
>
> We would like to clarify that ACTOR is distinct from ReFT in both design and motivation.
>
> 1. **Parametrization**: Rather than introducing **additional parameters**, ACTOR **fine-tunes a single layer** of the original model, avoiding the overhead associated with auxiliary interventions.
>
> 2. **Learning Objective**: Its objective departs from standard output-based losses by encouraging a **“just enough” shift** in the model’s internal representations, scaled by each query’s specific projection onto the refusal direction. This prevents both excessive and insufficient corrections, which can result when a one-size-fits-all approach is applied.
>
> 3. **Annotated Data**: Unlike ReFT, ACTOR does not rely on full response supervision as it draws on internal activation information as its supervision signal making it more **cost-effective** and straightforward to deploy.
>
> Motivationally, ACTOR is designed for cases where the degree of over-refusal varies by query, requiring individualized corrections (Sec 3.2). This focus drives both the training algorithm and the learning objective, ensuring a reduction in over-refusal while being computationally light. We will revise the appropriate sections to highlight these distinctions and justify our novelty.
>
>  ## (C)
>
> We would like to clarify that the **intent of the theoretical analysis** is **not to explain** why the “Just Enough” Shift preserves natural performance or why fine-tuning a single layer is sufficient. Rather, as stated in Lines 231–233, the theory is designed to offer **intuition behind the design of our training objective**—specifically, why subtracting out the projected activation shift with the refusal direction enables targeted correction for over-refusal, without imposing a uniform change across all inputs.
>
> We agree that theory is simplified to maintain clarity and provide a conceptual foundation for the method’s core ideas. It is **not intended to serve as a comprehensive performance guarantee.**
>
> As for the claims regarding natural performance preservation and single-layer sufficiency, we **support these empirically** through extensive experiments in Sec 4.2.
>
> 1. The “Just Enough” Shift mechanism maintains high performance on benign queries (Table 1)
>
> 2. Fine-tuning a single layer already yields substantial improvement in over-refusal mitigation without degrading natural performance (Table 3).
> We will revise the text to clarify the scope and role of the theoretical analysis and explicitly distinguish it from our empirical findings.
>
>  ## (D)
>
> **Comparison With Baselines**: Kindly refer to our rebuttal to Reviewer eEU9 (C)
>
> **Ablation Study on the Fine-Tuning Layer**: An ablation on the choice of the fine-tuning layer is present in Appendix C.1
>
> [1] Surgical, Cheap, and Flexible: Mitigating False Refusal in Language Models via Single Vector Ablation

---

> > ### Comment · Reviewer_Fyfs · 2025-04-02
> >
> > Thanks for the authors' response, which addresses most of my concerns. Though clarified by the authors, I still think the novelty and theoretical contributions are somewhat weak, but I appreciate the technical and empirical part of this work. Thus I have raised my score and am not opposed to acceptance.

---

> > > ### Author Response · Authors · 2025-04-06
> > >
> > > Dear Reviewer Fyfs,
> > >
> > > We are happy to hear that our responses have addressed your concerns and questions. We appreciate you taking the time to read our rebuttal and adjust your evaluation accordingly. We will incorporate all the clarifications, additional experimental results, and suggested modifications discussed during the rebuttal into our revised version. Thank you once again for your valuable, constructive feedback and for your consideration.
> > >
> > > Best regards,
> > >
> > > Authors

---

### Decision · Program_Chairs · 2025-05-01

**Decision:**

Accept (poster)

**Comment:**

ACTOR addresses the over-refusal issue in LLMs by finetuning a single layer based on activation patterns. It identifies and adjusts specific activation that triggers unnecessary refusals, while aiming to keep the model safe and useful. Reviewers found the method effective, simple and efficient. The method also can reduce over-refusal without hurting the model's safety and usefulness properties. There exist a few prior works on representation-based methods but the method is still unique. The paper mainly conducted experiments on single turn scenarios but could potentially be extended to multi-turn in the future work. The rebuttal addresses some concerns on baselines and ablations.